# Numerical and Experimental Comparison of Ducted and Non-Ducted Propellers

**Diego Villa *** , **Stefano Gaggero** , **Giorgio Tani** and **Michele Viviani**

Electrical, Electronics and Telecommunication Engineering and Naval Architecture Department (DITEN), University of Genova, 16126 Genova, Italy; stefano.gaggero@unige.it (S.G.); giorgio.tani@unige.it (G.T.); michele.viviani@unige.it (M.V.)
* Correspondence: diego.villa@unige.it; Tel.: +39-010-335-2345

**Abstract:** Ducted propellers are unconventional systems that are usually adopted for ship propulsion. These devices have recently been studied with medium-fidelity computational fluid dynamics code (based on the potential flow hypothesis) with promising results. However, these tools, even though they provide a good prediction of the forces and moments generated by the blades and the duct, are not able to provide insight into the flow field characteristics due to their crude flow approximations. On the contrary, modern high-fidelity viscous-based computational fluid dynamics codes could give a better description of the near and far-field flow of these particular devices. In the present paper, forces and the most significant features of the flow field around two ducted propellers are analyzed by means of both experimental and computational fluid dynamics approaches. In particular, accelerating and decelerating ducts are considered, and we demonstrate the ability of the adopted solver to accurately predict the performance and the flow field for both types. These results, in particular for the less-studied decelerating duct, designate CFD as a useful tool for reliable designs.

**Keywords:** Reynolds-averaged Navier–Stokes (RANS); adaptive mesh refinement (AMR); ducted propeller; validations and verifications

## 1. Introduction

Ducted propellers are one of the commonly used unconventional propulsion systems. They are designed to improve the propulsion performances, taking advantage of different physical principles depending on the shape of the adopted nozzle. In this regard, they are classified as propulsion systems operating with either an accelerating or decelerating duct (e.g., [1–6]).

The former is designed to increase the thrust provided by the propulsion system when a bollard pull condition (or, generally, a high-load condition) is considered. In this case, the duct, thanks to the accelerated flow provided by the propeller, develops a lift force pointing in the forward direction, which consequently adds up to the propeller thrust. This effect is magnified when a slow inflow field is considered (i.e., a bollard pull condition). This type of propulsion system is commonly adopted for working boats which require a significant propulsion force at slow speed (e.g., tugs).

In contrast, the decelerating duct's action on the flowfield is towards a reduction in the local velocity, causing an increased pressure region around the propeller (inside the duct). Higher pressures decrease the propeller cavitation inception index, ultimately reducing the cavitation extension on the blades and its undesirable effects (noise, vibration, erosion, etc.). This type of duct is suitable for fast, high-valued, ships where the occurrence of cavitation can have a strong negative impact on the entire ship's design and operability. The drawback of this system is represented by the consequent duct resistance, which must be compensated by the propeller itself.

Even if these types of propulsors have been widely adopted for several decades (mainly the accelerating ones), a thorough understanding of the relevant flow mechanisms is quite scarce in the literature since most of the previous research focused attention on design issues or on design methods [1,2,7–15]. Regarding the analysis of flow field peculiarities such as the vortex structures generated by the propeller, the available literature is even scarcer. Some pioneer analyses have been performed by [16–21], who experimentally and numerically analyzed the vortex structures and their interactions in the case of conventional propellers while only a few exceptions [22,23] deal with ducted systems.

In this context, the present study aimed to explore the capabilities of a viscous solver, namely *StarCCM+*, to address this particular problem. Section 2 summarizes the main parameters of the tested propellers and their reference functioning conditions. Section 3 provides an overview of the numerical method adopted, while in Section 4 the model-scale experiments carried out for the two propellers are described. After giving an overview of the numerical problem and the expected accuracy exploring the solution sensitivity to the grid size (Section 5), in Section 6 the uncertainty, from a numerical perspective, related to the setup of the computational domain and of the test case are investigated (i.e., open water or reverse open water setup). To gain better insight into the problem, the predicted propeller wake (i.e., velocity distributions on several transverse planes aft the propeller disk) is compared with the available measurements for two reference conditions, with and without the duct. Results of this comparison are summarized in Section 7. Finally, in Section 8 the vortex structures generated by the propellers, with and without the duct, are analyzed and discussed.

## 2. Propeller Test Cases

There is scarce literature data dealing with ducted propellers available, especially for those equipped with decelerating duct. Consequently, the present analyses were carried out considering geometries (both propeller blades and ducts) which were developed in the context of an industrial research project. In particular, they are two four-bladed, controllable pitch propellers equipped respectively with an accelerating and a decelerating nozzle. The accelerating ducted propeller has a 19A nozzle, while the decelerating ducted propeller uses a custom geometry. In both cases, the propulsive system was designed for its specific functioning condition in order to maximize the efficiency and to minimize the risk of cavitation, leading to two completely different propeller blade geometries. The decelerating duct was designed to maximize the internal pressure, increasing the inception margin and then reducing the blade cavitation at the design functioning condition by decelerating the flow field experienced by the propeller. In contrast, the accelerating duct was designed to increase the total thrust near the bollard pull condition. For the sake of clarity, we refer to the propeller designed to be coupled with the accelerating duct *propeller A*; we refer to the decelerating duct configuration as *propeller B*. A 3-D overview of the propellers and their nozzles is given in Figure 1. Their main geometric characteristics are summarized in Table 1.

**Table 1.** Main parameters of the propellers and ducts analyzed in this study.

| Propeller | A | B |
|---|---|---|
| Propeller type | CPP | CPP |
| Diameter (m) | 0.230 | 0.230 |
| $A_E/A_O$ | 0.689 | 0.725 |
| $P/D0.7R$ | 1.566 | 1.354 |
| # Blades | 4 | 4 |
| Duct Type | Accelerating | Decelerating |

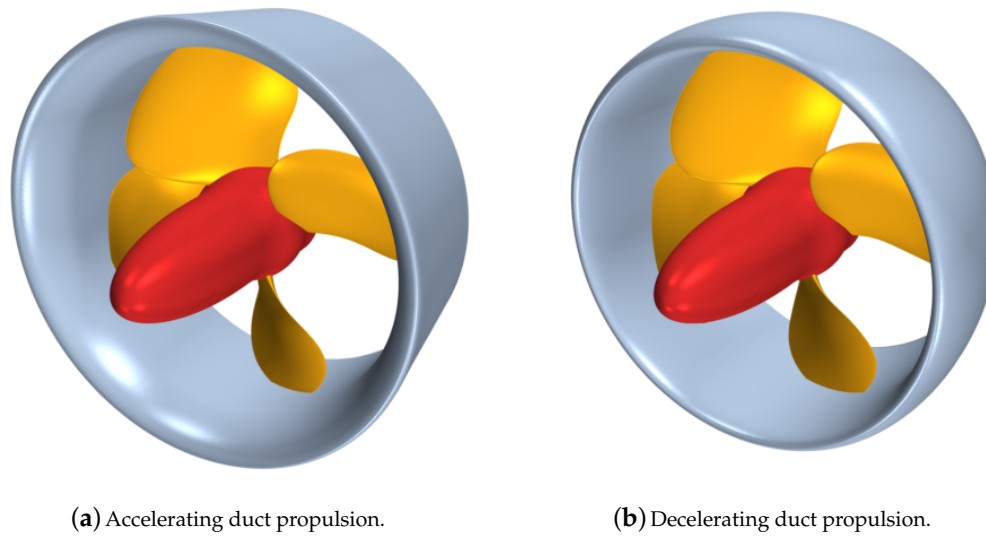

(**a**) Accelerating duct propulsion.      (**b**) Decelerating duct propulsion.

**Figure 1.** 3-D view of the propellers equipped with the ducts.

## 3. Numerical Methodology

The computational fluid dynamics (CFD) calculations proposed in this work were carried out with the commercial viscous flow solver *StarCCM+ V11*. *StarCCM+* is a general-purpose finite volume code widely used to solve naval problems. In the last several decades, it has been successfully applied to address several naval problems, ranging from simple cases such as the prediction of the hull resistance [24,25] or the numerical evaluation of the propeller performance [26], to complex interaction problems like rudder-propeller [27], self-propulsion estimations [28,29], or the prediction of ship maneuver characteristics [30]. Regarding applications directly related to the prediction of propeller performance, this code has demonstrated its accuracy in different conditions: with a spatial homogeneous wake [31] (i.e., in open water conditions), with inclined shaft [32], and in behind hull conditions, including cavitation [33]. Its reliability permitted its use as a tool devoted to the final verifications of the designs from an optimization process [11], or even to substiute the experimental campaigns for routine industrial applications.

Thanks to the multiple reference frame (MRF) approach and the propeller intrinsic geometrical periodicity, when open water conditions are considered (i.e., homogeneous inflow aligned with the propeller shaft), the flow problem can be conveniently re-written in a steady form. Equation (1) illustrates the steady version of the Reynolds-averaged Navier–Stokes (RANS) equations written in a rotating reference frame (non-inertial reference frame) in terms of absolute velocity: $U$ and $p$ stand for the velocity and pressure mean fields respectively, $\rho$ and $\nu$ are the physical characteristics (density and viscosity) of the fluid characteristics, $T^{Re}$ is the contribution to the mean flow field of the turbulent fluctuations, and $\rho(\omega \times U_a)$ is the rearranged Coriolis term. The subscripts $r$ and $a$ identify quantities evaluated respectively in the *relative* or in the *absolute* reference frame.

$$\begin{cases} \bigtriangledown \cdot U_a = 0 \\ \bigtriangledown \cdot (\rho U_r \otimes U_a) = - \bigtriangledown p + \nu \bigtriangleup U_a + T^{Re} - \rho(\omega \times U_a) + S \end{cases} \tag{1}$$

In this new form, the solution of RANS equations is several orders less computationally expensive compared to the unsteady version, mandatory if a fixed (inertial) reference frame is considered. Moreover, when vortex structures need to be predicted, the solver can easily take advantage of an adaptive mesh refinement (AMR) technique. This approach is straightforward when, as in present cases, the stationarity of the relative flow field is accounted, because the refinement procedure always involves the same domain regions This latest feature makes the recursive AMR process intrinsically convergent.

As usual, the turbulence has a primary importance for the reliable evaluation of the propeller performance (as shown in [34,35]). Current calculations rely on the *realizable k − ε* turbulence model, originally formulated by [36] and successively modified with the inclusion of a new transport equation for the turbulent dissipation rate [37]. This turbulence model satisfies some additional mathematical constraints on the normal stress, more consistent with the physics of the turbulence if compared to the original formulation. In addition, its implementation is able to solve both high- and low- Reynolds number flow problems, switching from a wall-solved boundary condition to wall functions modeling depending on the near-wall local velocities. This feature makes the solver more robust in terms of wall treatment.

The computational domain consists of a cylindrical domain characterized by a radius of 10 times the propeller diameter and a longitudinal extension upstream and downstream the propeller plane respectively of 5 and 10 times the diameter. This configuration, consistent with well-established guidelines, was preferred in order to minimize the influence of boundaries on the predicted flow in proximity of the propeller. Computational grids were realized using the tools provided directly in *StarCCM+*. Among the available, we preferred the hexa-dominant mesh generator, capable of including surface and volume refinements as well as refining cells based on field criteria. High-quality prism cells corresponding to the wall boundaries were also included to better capture the boundary layer gradients. As mentioned, the adaptive mesh refinement capabilities were also exploited in realizing the final computational grids. Compared to other refinement approaches, where the mesh is updated at each time step, the refinement process realized in this work adopts, starting from a reference mesh realized with usual local and volumetric refinements, a two-step procedure which refines the grid according to a user-defined volume field. These two steps are iterated until convergence is obtained (i.e., the new mesh is unchanged with respect to the previous one).

The first step consists of defining a volume field representing, over the entire computational domain, the desired cell size. This field is a function of some values of the computed flow field (e.g., the intensity of the vortical structures) which are considered representative of the regions deserving refinement, and it is first calculated for the reference mesh, and then at the end of each refinement iteration. Current calculations make use of the Q criterion to select regions for successive grid refinements.

The second step instead realizes the mesh including the local refinements identified by the field function of the first step, interpolates the solution computed using the previous grid on the current grid, and runs the calculation until residuals decrease under a certain threshold (or for a given number of iterations, whichever is smaller).

## 4. Experimental Methodology

The hydrodynamic characteristics of the propellers studied in the present work were also investigated through dedicated experimental campaigns. In particular, experimental data included propeller open water characteristics for both propellers and, for the decelerating ducted propeller only, detailed flow measurements. Open water tests were carried out in the towing tank at the SVA (Schiffbau-Versuchsanstalt Potsdam) in accordance with ITTC procedure. Propeller characteristic curves for the ducted configuration, hence including forces acting on the duct, were measured. Experiments without the duct were considered as well. These tests were carried out adopting the pulling propeller configuration. In order to verify the effect of the Reynolds number, multiple tests considering different values of the shaft rate (10, 12.5, and 15 RPS) were considered. Numerical calculations were compared to experiments measured at the higher propeller revolution rate to reduce the influence of the laminar boundary layer by promoting earlier boundary layer transition. Further experimental investigations were carried out at the cavitation tunnel of the University of Genova. The facility is a Kempf and Remmers medium-size closed water circuit tunnel, with a test section of 0.57 m × 0.57 m × 2 m. This second campaign included cavitation observations and flow measurements by means of laser Doppler velocimetry (LDV). The equipment adopted for the flow surveys was a four-beam, two-color fibre optic LDV system with back-scatter collection optics

(Dantec Fiber Flow). The light source was a 5 W argon ion laser operating at 514.5 nm (green) and 488 nm (blue). In order to solve the zero velocity ambiguity and to reduce angle bias, a 40 MHz Bragg cell was used. The optical transducer head had a focal length of 400 mm and a beam separation of 38 mm. The measuring volume was an ellipsoid volume whose minor and major diameters were 190 μm and 4 mm, respectively. The probe assembly was mounted on a three-axis computer-controlled probe traversing mechanism whose minimum linear translation step was 8 μm. Measurements were analyzed by means of an ensemble averaging technique. To obtain statistically accurate ensemble averages, a total of 200,000 validated samples were acquired at each measuring position, for each velocity component. Instantaneous values were sorted into 360 phase bins, exploiting the knowledge of the arrival time of the sample and the synchronization signal generated by the propeller shaft encoder. Each bin represented a particular angular position among a total of 360. A comprehensive review of errors in laser Doppler velocimetry measurements and guidelines to evaluate them can be found in [38–41]. The statistical uncertainty in mean and rms velocities depends on the number of sampled data, turbulence intensity, and confidence level. Considering a confidence level of 95%, a local turbulence intensity of 20%, and a minimum number of samples of 150 per bin, uncertainties of ±3% and ±7% are expected for the mean and rms velocities, respectively. Flow measurements were carried out for eight axial stations considering both upstream and downstream positions, as reported in Figure 2. Stations 5 and 6 were defined with small axial distance in order to allow computation of the vorticity between them. Flow surveys were carried out for the propeller with and without duct. The propeller operational condition under study was the design condition for the ducted configuration: the shaft rate was kept constant and equal to 18 RPS, the advance speed was adjusted to obtain the design propeller thrust coefficient. Similarly, the operational condition for the propeller without duct was defined using the same shaft rate and the identity of the propeller thrust coefficient.

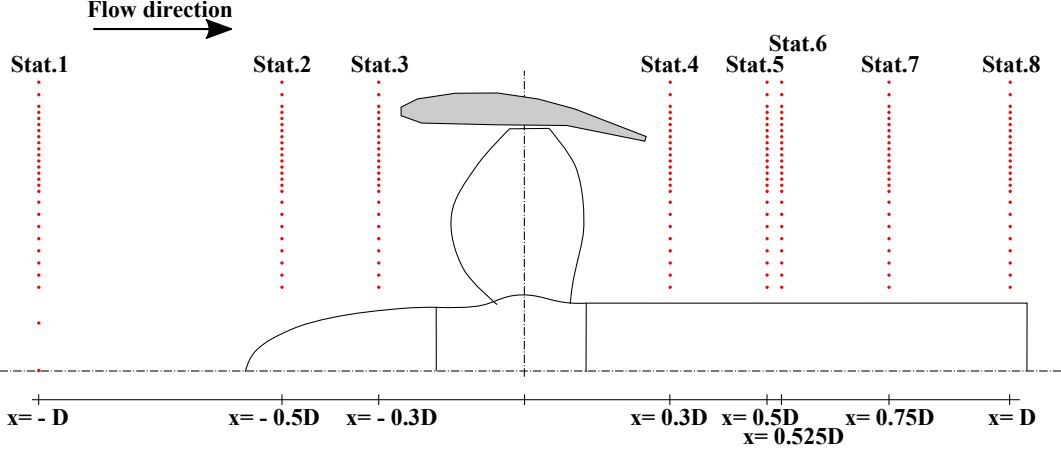

**Figure 2.** Experimental measurement section positions.

## 5. Validation and Verification of the Global Forces

Thanks to the available experimental data, a preliminary comparison of the CFD predictions with the measurements was possible. For both systems, two configurations were considered: propeller with and without the duct. In all cases, calculations were carried out with the reference mesh (the one before any adaptive mesh refinement). Only standard refinements that were local at the leading and trailing edges of blades and nozzles and zonal around the propulsor were included, leading to a computational grid consisting of about 1.5 million cells for a single blade passage.

Figures 3 and 4 show the non-dimensional forces and moments developed by the propulsion systems at different functioning conditions. In particular, for the configuration without the duct, the thrust ($K_T$) and torque ($K_Q$) coefficients are plotted against the advance coefficient ($J$). When a

ducted configuration was considered, the plots also include the thrust (or resistance, depending on the functioning point) provided by the duct and the propeller separately. These coefficients were made non-dimensional following the well-known formulations reported in Equation (2):

$$J = \frac{V_a}{nD} \quad \text{and} \quad K_T = \frac{T}{\rho n^2 D^4} \quad \text{and} \quad K_Q = \frac{Q}{\rho n^2 D^5} \tag{2}$$

where $V_a$ is the mean inflow to the propeller with a prescribed density $\rho$. $D$ and $n$ are respectively the propeller diameter and the rate of revolution of measurements or computations. Since all the geometries for these analyses were provided under a confidentiality agreement, all the results were further divided by a reference value.

Only a small range of values across the design advance coefficient was considered in calculations. The overall accuracy was satisfactory, similar to what is usually reported for this type of simulation. In general, trends of thrust and torque were accurately predicted for all the configurations, with or without the nozzle, even if some discrepancies were noted when off-design conditions were considered. This aspect was magnified in the case of ducted configurations, for which there were larger discrepancies, in particular regarding the delivered thrust and the absorbed torque of the blades, whereas duct forces were predicted with a satisfactory accuracy. The better agreement observed in the case of the total delivered thrust is a result of errors on propellers and ducts forces cancelling each other.

To explore the influence of the grid density on the computed forces and moments, the Richardson extrapolation [42] was used to assess the convergence trend. Results of the analyses are proposed for both the configurations, with and without the nozzle, using in addition to the reference mesh, a coarser (about 900,000 cells when the nozzle was included) and a finer mesh (about 2.4 million cells, with nozzle) shown in Figure 5. Each mesh was obtained by changing the global mesh size and leaving the same refinement structures.

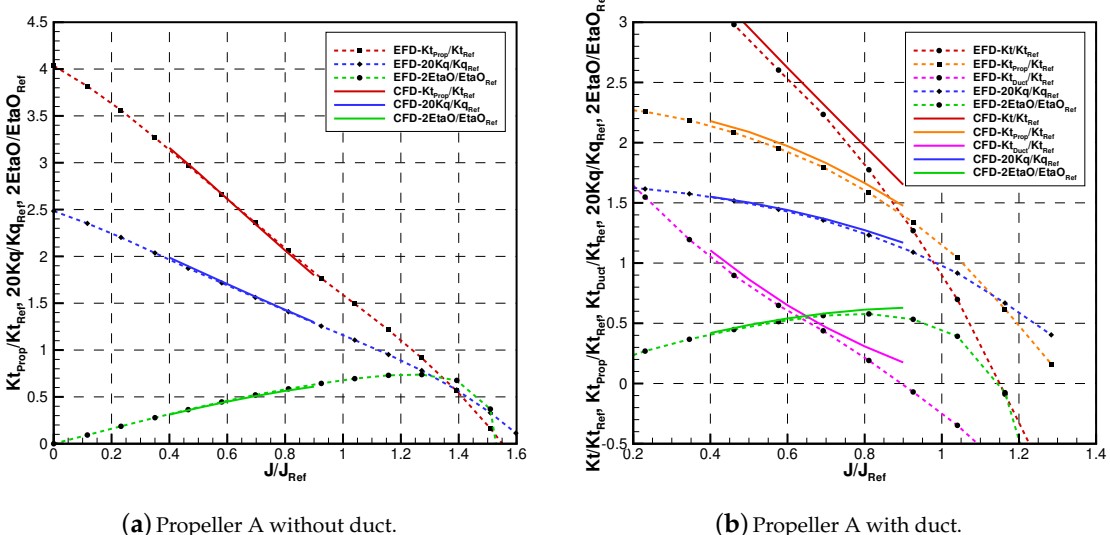

(**a**) Propeller A without duct.  (**b**) Propeller A with duct.

**Figure 3.** Open water tests for the accelerating ducted propeller: computational fluid dynamics (CFD; solid lines) vs. experimental fluid dynamics (EFD; dashed lines).

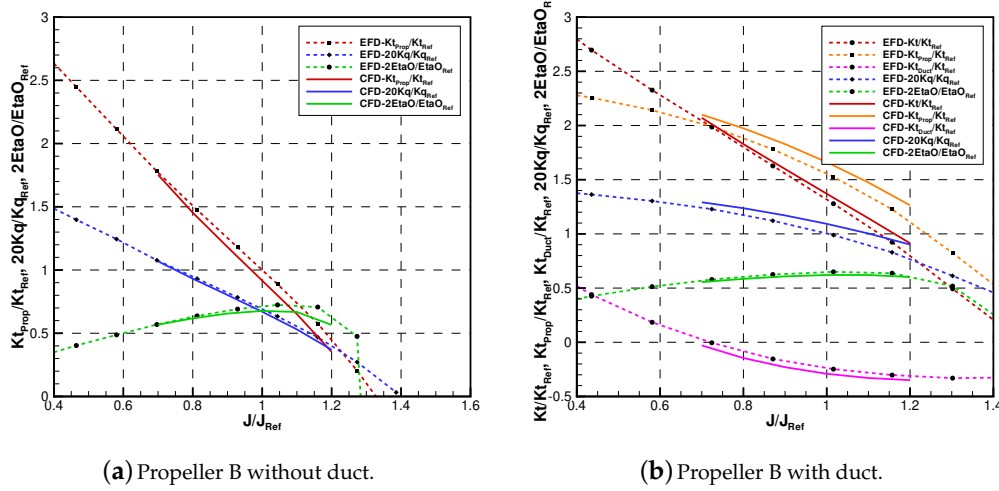

(**a**) Propeller B without duct.

(**b**) Propeller B with duct.

**Figure 4.** Open water tests for the decelerating ducted propeller: CFD (solid lines) vs. EFD (dashed lines).

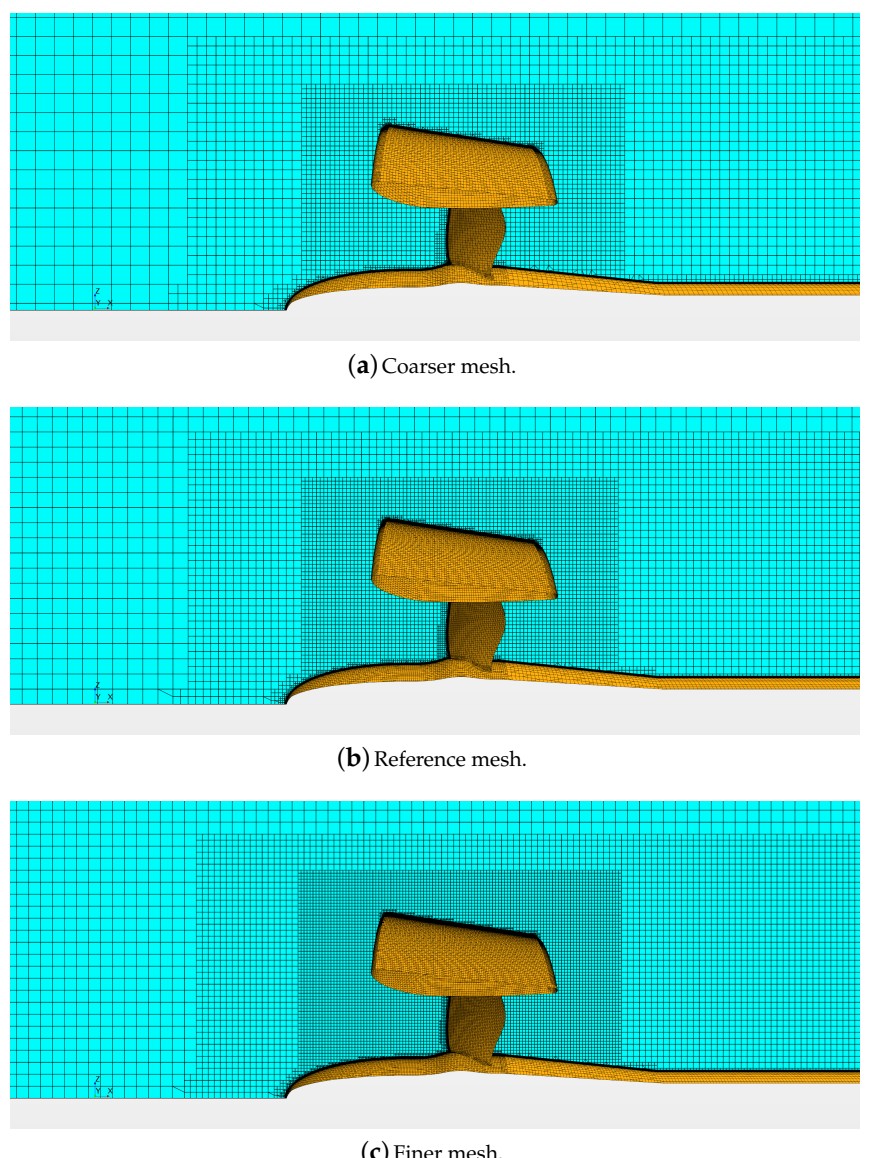

(**a**) Coarser mesh.

(**b**) Reference mesh.

(**c**) Finer mesh.

**Figure 5.** Sketches of the adopted meshes for the sensitivity analysis.

Table 2 reports the percentage of the total thrust variations with respect to the extrapolated values obtained by the Richardson procedure. Doubling the mesh size (from the reference to the finer one) had only a small impact on the predicted thrust. This is consistent with the uncertainty evaluated in similar calculations [33] and workshops [43]. Considering that the results of the reference mesh differed from the extrapolated values by less than 2%, this mesh arrangement can be considered accurate enough to be adopted in further analyses. In any case, it is also important to point out that when a finer mesh was used, the numerical discrepancy with respect to the extrapolated value was further reduced to less than 0.5%. This was the case for the calculations with the AMR which, at the end of the iterative refinement, also provide substantially denser grids near the propeller blades and near the nozzles.

**Table 2.** Percentage variation of the total delivered thrust with respect to the extrapolated value.

| Case | Coarse | Reference | Fine |
|------|--------|-----------|------|
| Propeller A without duct | 0.19 | 0.50 | 0.19 |
| Propeller A with duct | 1.41 | 1.09 | 0.61 |
| Propeller B without duct | 3.17 | 2.66 | 0.64 |
| Propeller B with duct | 2.72 | 0.46 | 0.08 |

## 6. Analysis of the Test Configuration

All the comparisons of the previous section considered CFD calculations realized using the same "open water" configuration adopted during measurements at the towing tank. This means that, for instance, the propeller was positioned in front of the shaft (pulling configuration), and a streamlined boss cap was used. Some additional simulations were performed modifying the propeller configuration (the so-called reverse propeller open water configuration) to explore its effects on the predicted performances, at least from the numerical point of view. In particular, this analysis focused on the shaft direction and on the boss cap shape.

During towing tank tests, propellers are commonly positioned in front the shaft (to reduce the in-flow disturbances), while when the propeller is in a behind-hull configuration, the propeller shaft is positioned in front of the propeller (pushing configuration). In light of this, additional simulations were performed reversing the shaft direction for one of the two propellers, both with and without the duct. Calculations always considered a shaft line extending from the propeller to the inlet (pushing configuration) or to the outlet (pulling configuration) of the computational domain.

Figure 6 shows the percentage variation of the predicted total thrust and torque at different propeller loads. The results show that when the propeller was highly loaded (low value of *J*), the effect of the shaft direction on the thrust became non-negligible (up to 6%). This effect was almost the same for both cases, with and without the nozzle.

Figure 7 shows the comparison, on a longitudinal section of the computational domain, of the non-dimensional pressure field computed for the two shaft configurations (propeller without duct). The near-field radial pressure distributions are quite similar, with only differences seen at the root of the blades. These discrepancies are due to the different shaft geometries causing differences in the characteristics of the propeller inflow and of its wake field: considering the pushing configuration, the inflow includes the presence of the boundary layer of the shaft, whereas a hub vortex characterizes the propeller wake. Instead, for the pulling configuration, the inflow is only influenced by the presence of a stagnation point on the boss cap while no hub vortex is present in the wake of the propeller. Moreover, the presence of the shaft slightly reduces the cross section of the propeller slipstream. These modifications of the inflow slightly alter the pressure distributions over the blades, definitely determining the variations of the propeller load seen in Figure 6. For the sake of completeness, further experimental measurements were necessary to confirm these observations, which needed to be accounted for in the setup of numerical analyses to make them suitable for fair comparisons with experiments.

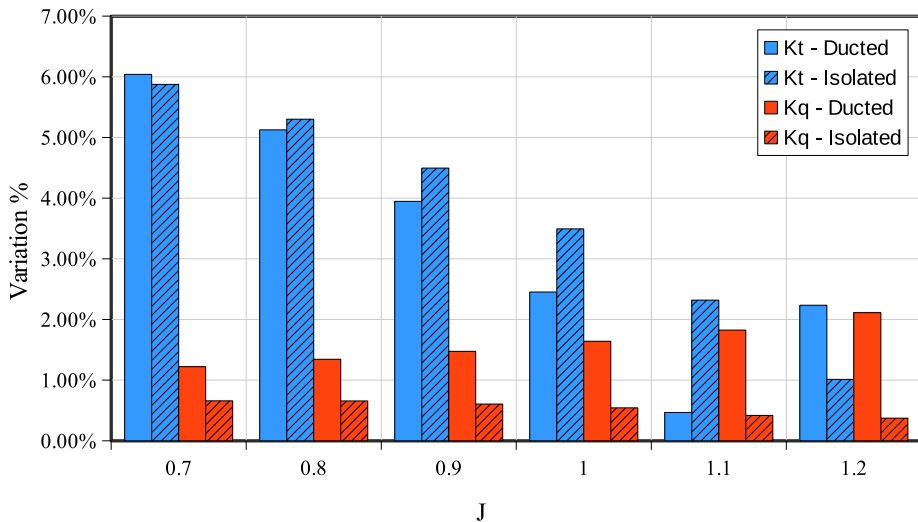

**Figure 6.** Percentage variation (pushing with respect to pulling condition) of the thrust and torque coefficients varying the shaft direction.

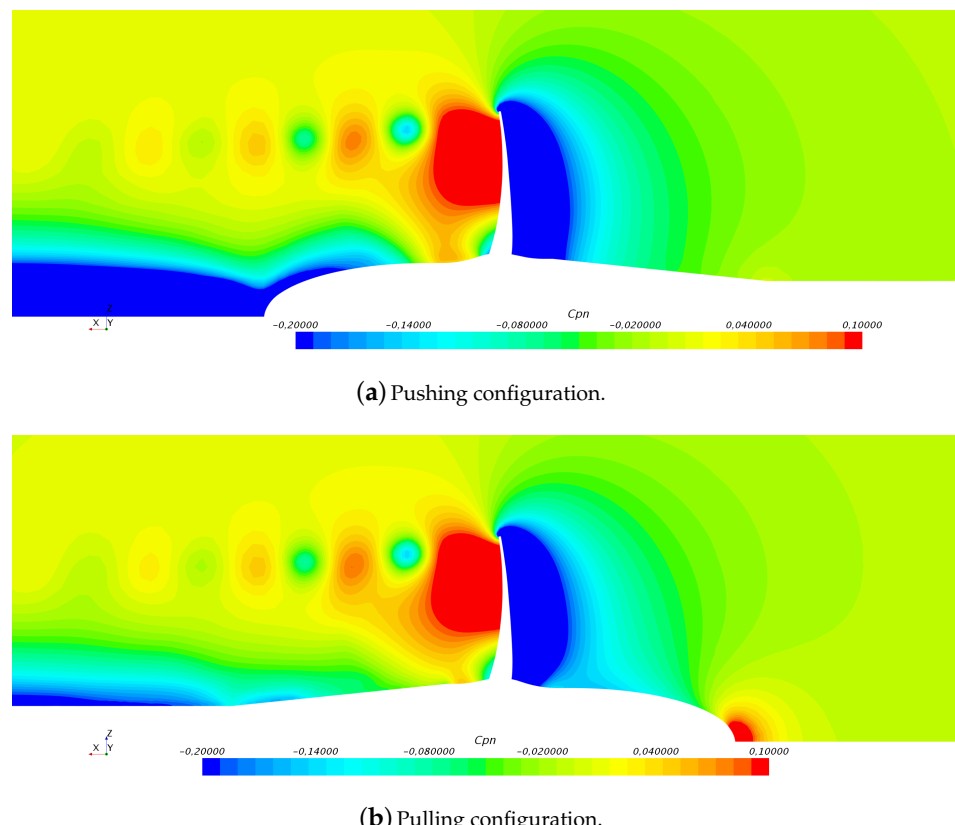

(**a**) Pushing configuration.

(**b**) Pulling configuration.

**Figure 7.** Shaft direction effect on pressure field at $J = 0.7$.

Two different caps were considered (see Figure 8) to investigate the role of the boss cap shape in predicted propeller performances. The first (*Model hub*) was the cap commonly adopted for model tests when the pulling configuration is considered. It has an elliptical shape with smooth geometry variations to avoid separation effects that could alter the propeller inflow. The second (*Ship hub*) was the usual boss cap employed in behind-hull functioning for reverse propeller open water tests (pushing configuration). It is designed to limit the merging of blade root vortexes in a single stronger hub vortex, mitigating its intensity to avoid (or at least reduce) root vortex cavitation issues. Since this shape is

appropriate only for pushing functioning, the comparison of Figure 9 was carried out with the same shaft configuration (i.e., shaft upstream the propeller) closed, downstream, by the (*Model hub*) or by the (*Ship hub*). Results show opposite trends with respect to those observed for the shaft direction, since the effect of the boss cap shape was higher for lower loads (high values of *J*). Discrepancies were magnified because the graph reports the relative variations; in fact, the effect of the boss cap geometry on the absolute values of thrust and torque was practically constant and almost negligible for any loading conditions. Therefore, since the aim of the computations is to assess the characteristics of the propulsion system in terms of thrust, torque, and efficiency, the effect of the boss cap geometry can be reasonably neglected.

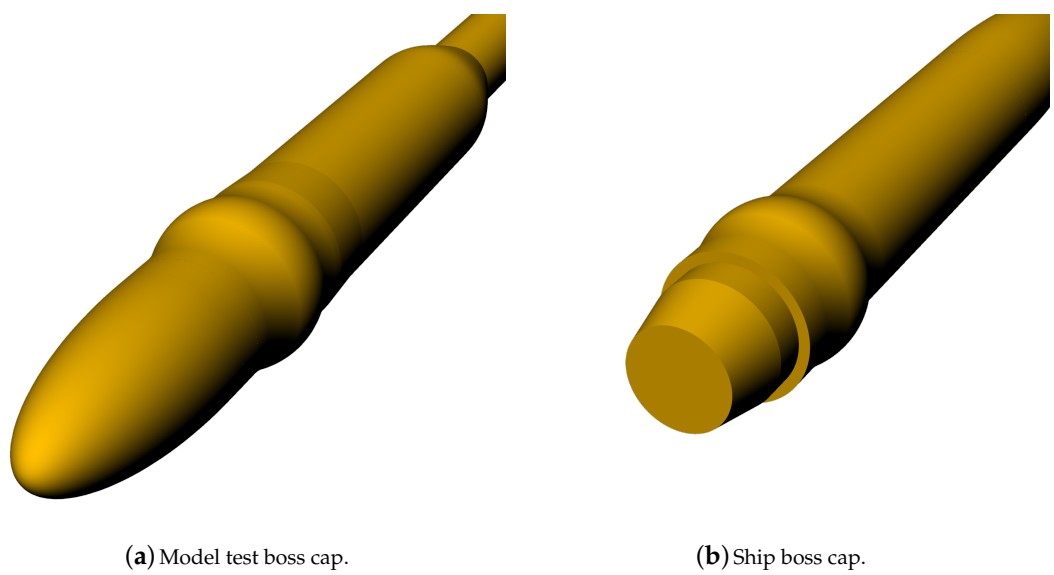

(**a**) Model test boss cap.　　　　　　　　　　　　(**b**) Ship boss cap.

**Figure 8.** Numerically tested hub shapes.

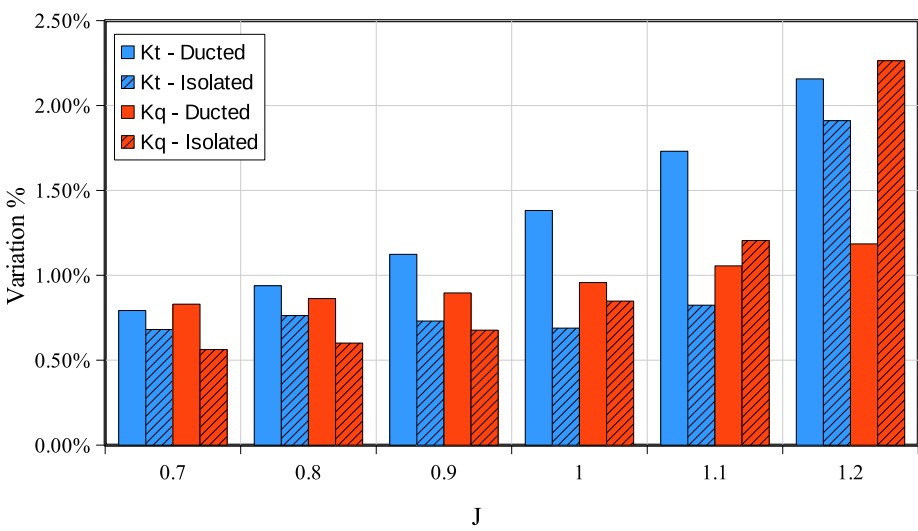

**Figure 9.** Percentage variation (ship boss cap with respect to model test boss cap) of the thrust and torque coefficients varying the boss cap shape.

## 7. Comparison with Experimental Flow Measurements

After the preliminary validation of calculations in terms only of global propeller characteristics, and after some consideration of the effects of different test configurations on the predicted forces, the flow field was analyzed by comparing numerical results with the measurements carried out at the cavitation tunnel. The aim was to assess the capabilities of the numerical approach to characterize the



main feature of the flow field in the complex configurations represented by ducted propellers, analyze the interactions between the blade and the nozzle in the gap between the blade tip and the duct, and finally, to gain a better insight into the flow characteristics of these propulsors.

In contrast to open water calculations, all the calculations carried out for the comparisons proposed in this section took advantage of the adaptive mesh refinement technique. The AMR was necessary to realize sufficiently fine and localized meshes to accurately trace, by limiting the numerical dissipation, the vortical structures generated by the propulsor also far from the blades, without increasing the total cell count by too much, while also keeping the computational time reasonable for the daily applications needed in the preliminary design phase. This approach, applied to the propeller flow field, has already been successfully explored in [44]. Compared to previous analyses (about 1.5 million cells for ducted propellers' open water analyses), final grids realized with the AMR had about only 7 million cells (2.7 million for the configurations without the nozzle) providing, at the same time, a mesh that was three times finer in correspondence with and around the predicted vortical structures. A bulk refinement of the propeller wake to ensure the same mesh density realized by the AMR process, regardless of the vortexes positions, would have led to more than 100 million cells, making the analysis unaffordable.

Figure 10 shows, on a longitudinal section of the computational domain, the field criterion used to refine the volume mesh in the propeller wake region in correspondence of the final converged step of the iterative AMR process. The colors represent the four different refinement levels forced locally on the mesh: 0 (blue) stands for the unrefined region, 1 (cyan) and 2 (light green) for intermediate refinements, and 3 (red) for the most refined one. It is clear that for all the configurations, the hub region was strongly refined due to the presence of root vortexes. Nozzle wakes, influenced by the tip vortex from the blades, generate complex interactions. Without the duct, the tip vortex region is instead refined only in the core of vortexes. With such a complex wake shape, it is evident that it would be extremely challenging to keep the total cell count low for a classical end-user approach (selection of the refined region in the pre-processing stage directly by the user).

Experimental flow measurements were available only for the propeller with decelerating duct. Therefore, the comparison between CFD and EFD was limited to this propulsion system, with and without the duct. To deeply analyze the flow field behavior, four transverse sections (normal to the propeller shaft) were compared: one in front of the propeller at a distance of 0.3 x/D with respect to the propeller plane and the other three respectively at 0.3, 0.5, and 0.75 x/D downstream.

Figure 11 shows the non-dimensional axial velocity component with respect to the undisturbed inflow to the propeller for the configuration without the duct. The flow field behavior was well predicted compared to the measurements, except for the tip and root regions, where some small discrepancies are highlighted.

This effect is furthermore highlighted in Figure 12, where the axial velocity was circumferentially averaged. The comparison between the measured mean values and the predicted mean values shows that CFD strongly overestimated the tip blade load, resulting in a higher flow velocity in correspondence of the outer radial positions. On the other hand, lower velocities were observed at inner locations. This is not surprising, since computations were carried out imposing the total thrust, hence the higher load of propeller tip was compensated by the lower load close to the blade root. This discrepancy is not very well documented in the literature for other propeller geometries, because of the few available flow measurements. Further comparisons are needed to better understand if the problem can be ascribed to the adopted numerical approach or if it is connected to this particular propeller geometry. Notwithstanding the presence of this slight shift in the velocity distributions, the position of the tip vortex was correctly predicted for all the sections downstream, demonstrating that the local wake pitch also had a reasonably good agreement with the measurements. Instead, a significant under-prediction of the vortex intensity (given by the velocity intensity) was observed when comparing, in Figure 13, the radial and tangential velocities (made non-dimensional with respect

to the undisturbed flow velocity) with the LDV data. The position and the shape trailing vortical wake were instead reasonably predicted.

When the ducted configuration was considered (Figure 14), a small velocity shift between computations and measurements was observed as well. In this case, the analysis of the circumferentially averaged velocities of Figure 15 better point out the differences.

Additionally, for this case, the velocity reduction in the wake of the nozzle was significantly over-predicted, probably ascribable to a insufficiently accurate modeling of its boundary layer. Indeed, the separation and the backflow at the trailing edge of the duct pose several numerical difficulties: reliable prediction, for instance, may require fully resolved boundary layer calculations, which are computationally expensive and beyond the scope of current analyses using wall function models.

Nevertheless, even if there was an overall discrepancy in terms of shift between predicted and measured axial velocity distributions, in the presence of the nozzle the blade wake and the tip vortices were quite well predicted—especially in terms of positions. As already observed in the case of the propeller without the duct, their intensities (especially for the tip vortex), were under-predicted (Figure 16).

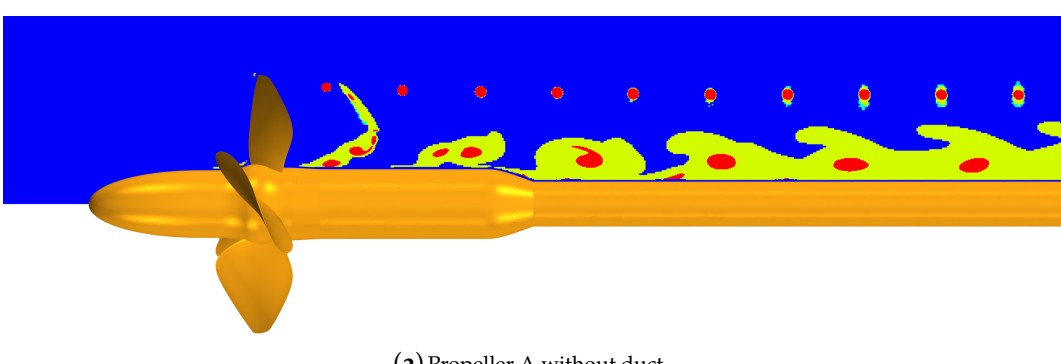

(**a**) Propeller A without duct.

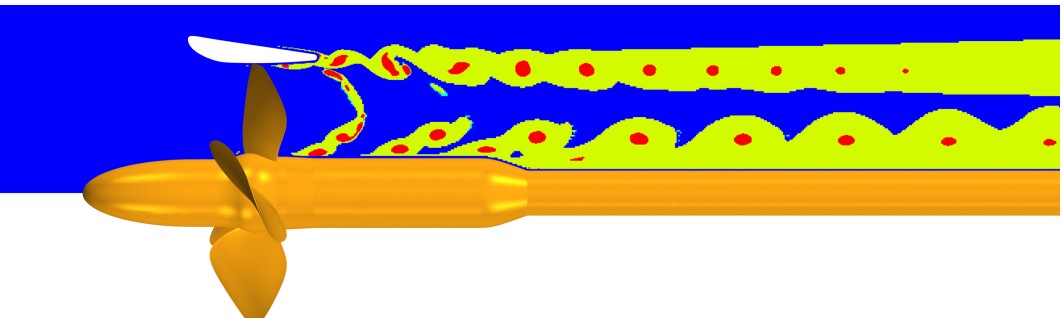

(**b**) Propeller A with duct.

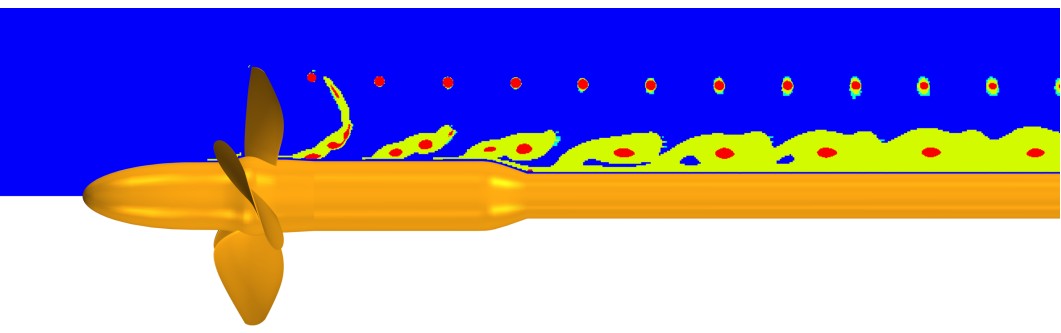

(**c**) Propeller B without duct.

**Figure 10.** *Cont.*

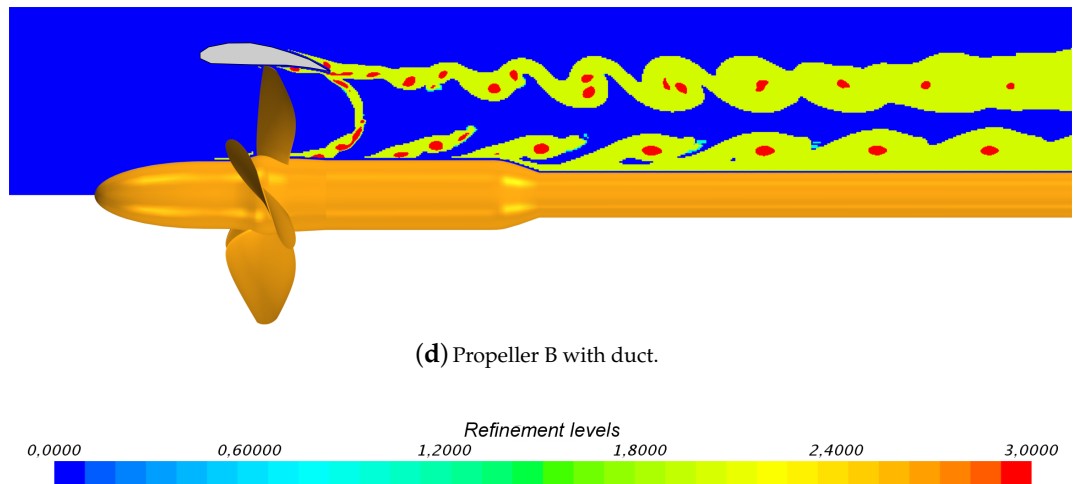

(**d**) Propeller B with duct.

**Figure 10.** Longitudinal section of the obtained refinement levels.

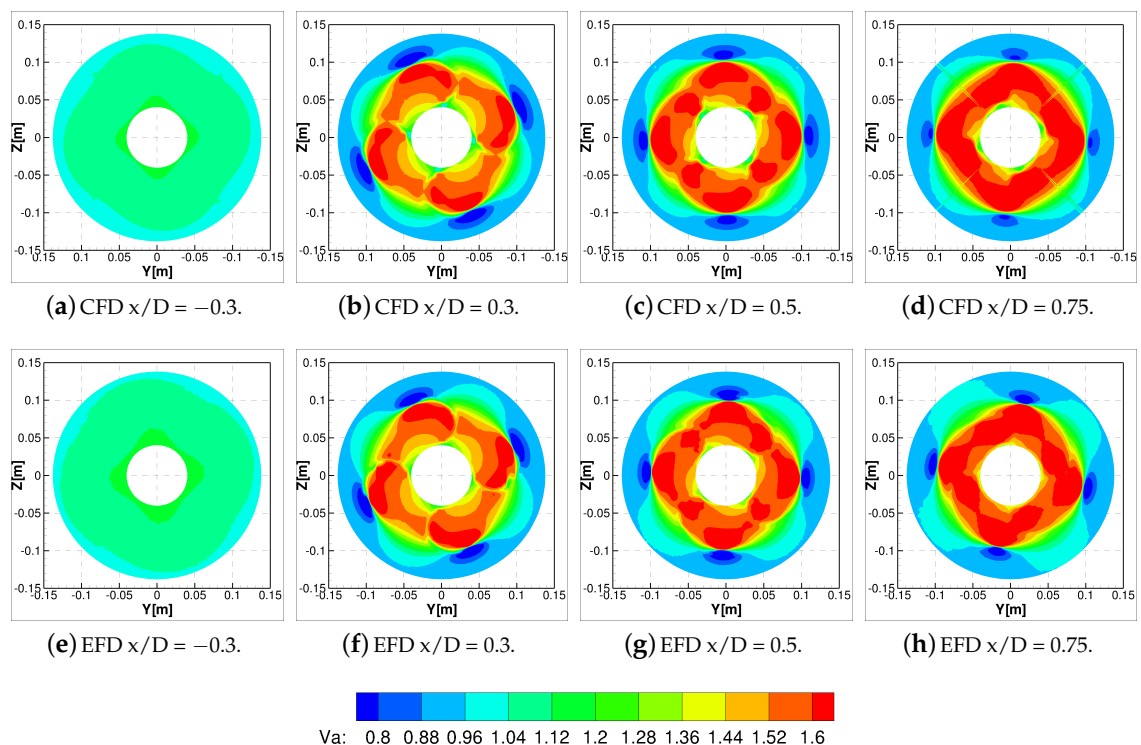

**Figure 11.** Axial non-dimensional velocity at different transversal sections for no-duct configuration.

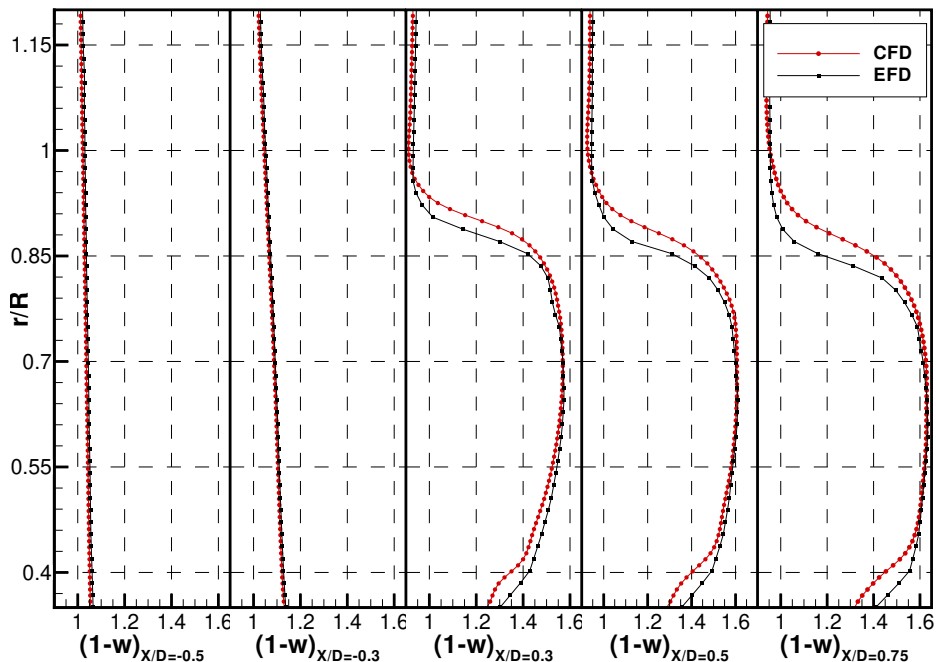

**Figure 12.** Circumferential axial velocity average for no-duct configuration.

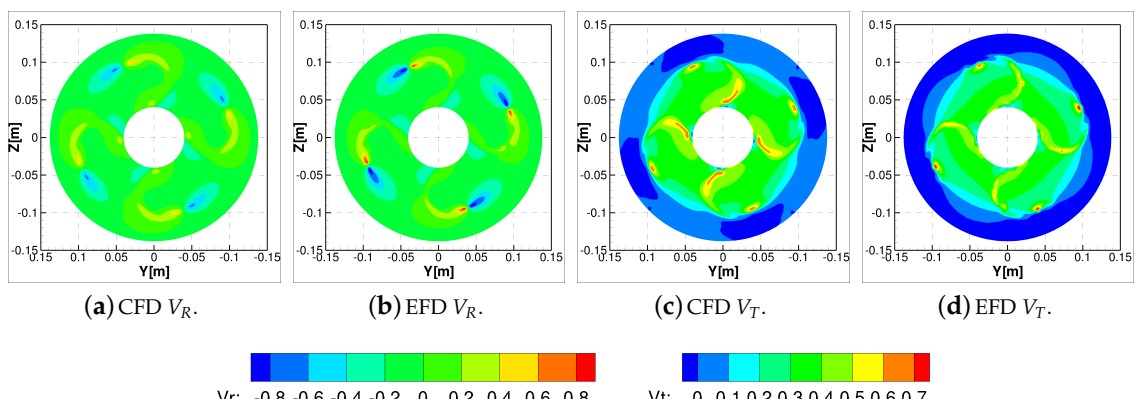

(**a**) CFD $V_R$.　　(**b**) EFD $V_R$.　　(**c**) CFD $V_T$.　　(**d**) EFD $V_T$.

**Figure 13.** Radial ($V_R$) and tangential ($V_T$) non-dimensional velocity at 0.3 x/D transversal section without duct.

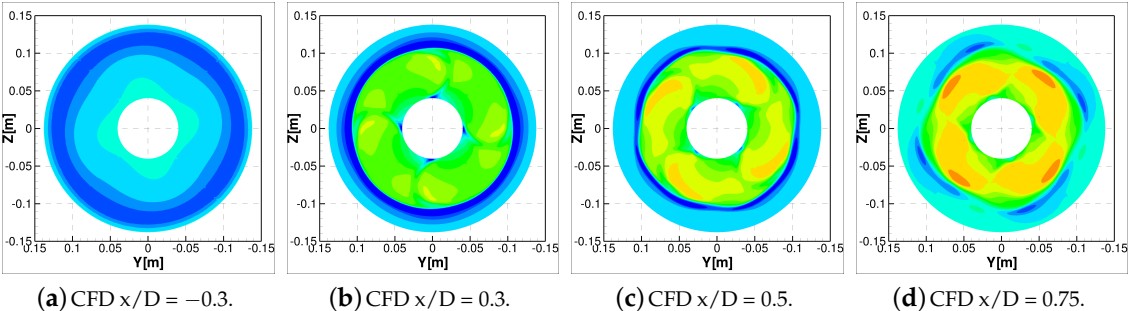

(**a**) CFD x/D = −0.3.　　(**b**) CFD x/D = 0.3.　　(**c**) CFD x/D = 0.5.　　(**d**) CFD x/D = 0.75.

**Figure 14.** *Cont.*

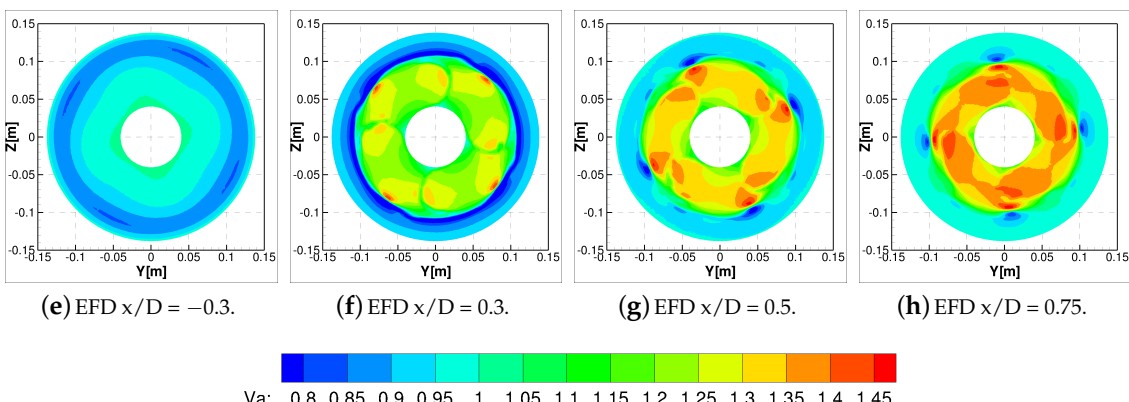

Figure 14. Axial non-dimensional velocity at different transversal sections for the ducted configuration.

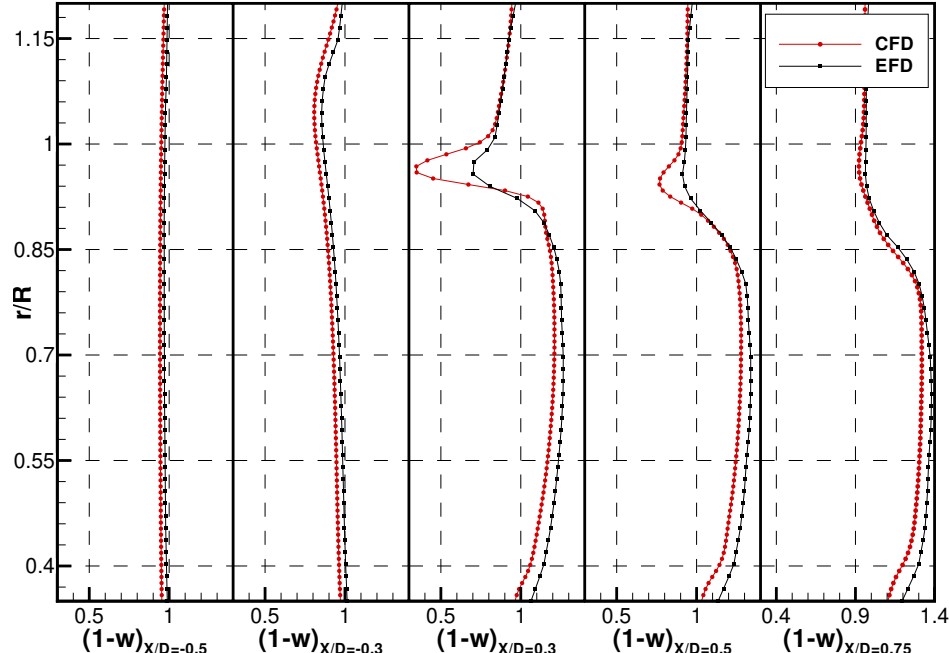

Figure 15. Circumferential axial velocity average for the ducted configuration.

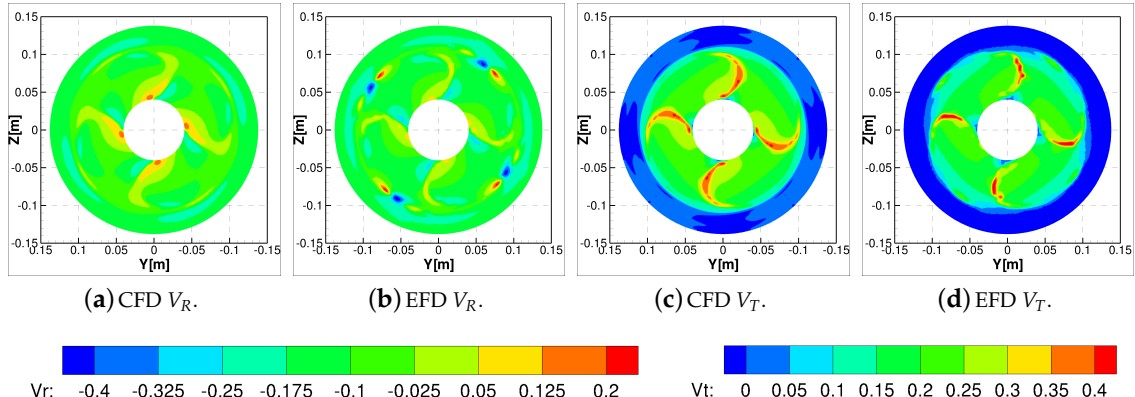

Figure 16. Radial ($V_R$) and tangential ($V_T$) non-dimensional velocity at 0.3 x/D transversal section with duct.

## 8. Flow Field Analysis

The vortical structures of the considered configurations were finally analyzed using the description of the flow available from data gathered by CFD simulations to deeply investigate the physics of the problem. The Q-criterion field, which was already employed in the iterative adaptive mesh refinement, was also employed to visualize and track, through the computational domain, the vortical structures generated by the blades and the ducts of the propulsors. The Q criterion represents the second invariant of the velocity gradient field, as defined in Equation (3) by [45].

$$Q = \frac{1}{2}(\parallel \Omega \parallel^2 - \parallel D \parallel^2) \tag{3}$$

where $D$ and $\Omega$ are respectively the symmetric and anti-symmetric parts of the velocity gradient. Positive values represent regions where the vorticity magnitude is higher than the rate of strain.

Focusing the attention on the vortical structures, Figure 17 reports the iso-surfaces of the Q-criterion field set equal to 1000 s$^{-2}$. Tip vortexes and, more generally, vortical structures from the propeller/nozzle interactions, were accurately identified in the computational domain regardless of the propulsive configuration under investigation. The AMR showed its influence exactly in these cases. Without the nozzle it was possible to observe strong vortices, extended over the entire computational domain. The presence of the nozzle, and the turbulent viscosity shed in the wake (which is generally overestimated in case of RANS calculations), contributed to a reduction of the strength of the blade tip vortexes as a result of their interaction with the viscous wake of the nozzle. However, the interaction itself is responsible for much more complex vortical structures which now include two rolled up vortexes.

Exploring in detail the vortical structures generated by the two blades without the duct (Figure 18), only a single, strong tip vortex (traced by the green lines in the figure) was observed. The two propellers in this analysis developed the same load they respectively had in correspondence of their design points with the duct. When the duct was included in the analyses, both propellers still showed a strong tip vortex having a pitch substantially unchanged with respect to the calculations at the same load—but without the nozzle—of the previous Figure 18. Small differences can be appreciated, since the influence of the ducts in terms of modified velocity fields cannot be completely described only by the thrust identity. However, this strong tip vortex is part of a more complex vortical structure (Figure 19) generated by the interaction with the nozzle.

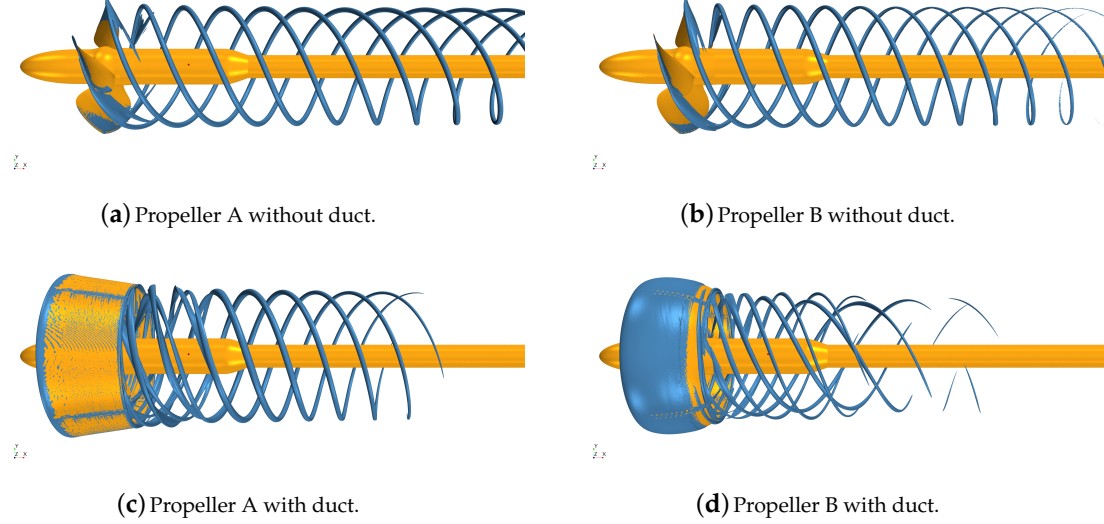

(**a**) Propeller A without duct.

(**b**) Propeller B without duct.

(**c**) Propeller A with duct.

(**d**) Propeller B with duct.

**Figure 17.** 3-D view of computed vortexes tracked by the Q criterion.

In addition to the vortex from the tip of the blade, a second vortex structure (traced by the magenta lines in Figure 19) starting from the leading edge of the blade tip can be observed. The pitch of this secondary vortex was almost half that of the primary vortex from the tip. This new structure is generated by the flow interactions in the gap between the blade and the duct that, for these propellers, is equal to 2% of the propeller diameter. This secondary vortical structure has the feature of the so-called tip leakage vortex, experimentally observed in ([22]). Additionally, in that case, the leakage vortex had a pitch of about half that of the tip vortex. These two vortexes do not interact each other until they reach the trailing edge of the duct, where they are influenced by the wake of the duct. This interaction mainly produces the pitch variation of the leakage vortex (dashed magenta line), which assumes the same pitch of the tip blade vortex (red line). Due to the presence of two co-rotating vortical structures, a roll-up phenomenon occurs in the propeller wake between the two vortexes. The mutual positions start to rotate, because each vortex experiences the induced velocity of the other. In addition, they also start to approach each other, as better shown in Figures 10b and 11d, until they merge into a single vortex.

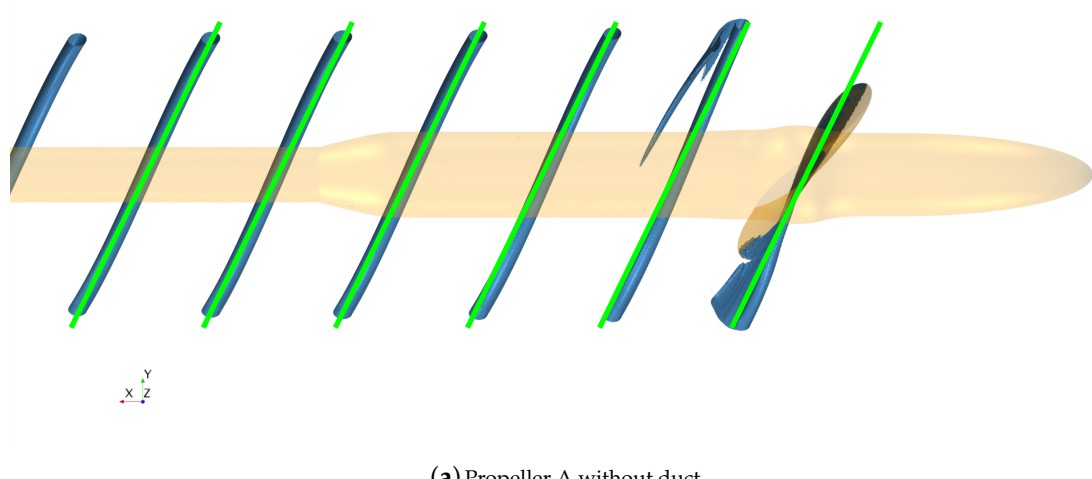

(**a**) Propeller A without duct.

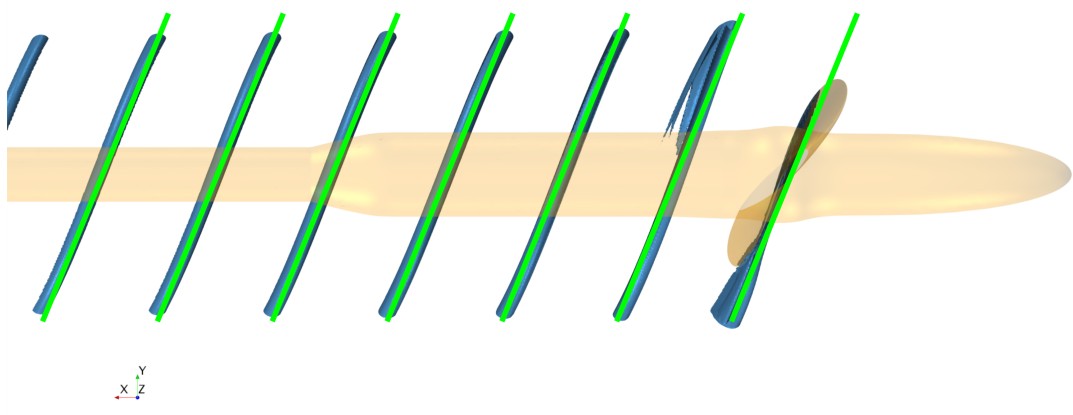

(**b**) Propeller B without duct.

**Figure 18.** Qualitative analyses of the vortex structures generated by the ductless propellers.

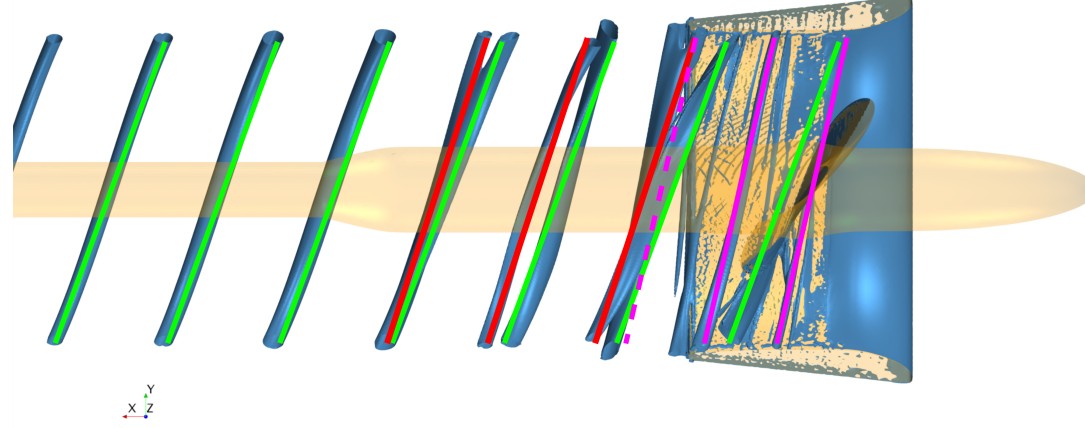

(**a**) Propeller A with duct.

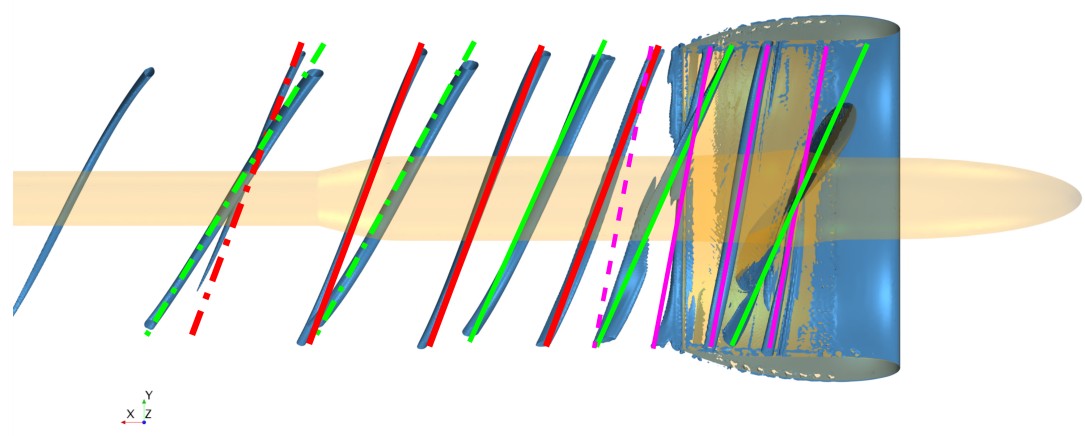

(**b**) Propeller B with duct.

**Figure 19.** Qualitative analyses of the vortex structures generated by the ducted propellers.

These considerations underline how the presence of the duct increases the complexity of the flow with possible effects on the propeller side effects, such as noise and vibrations. In fact, the presence of multiple vortical structures may influence the radiated noise in different ways. The interaction among multiple vortexes is expected to affect the stability of the vortexes themselves and of the entire propeller wake. Wake instabilities are known to significantly influence propeller radiated noise. Similarly, vortexes can have an impact on objects present downstream the propeller, such as rudders. Finally, the interaction between the two main vortical structures, especially for the propeller equipped with a decelerating duct, is known to reduce the strength and the stability of the main tip, vortex with major effects on cavitation (if present), and on the related noise.

These kind of phenomena are not directly analyzed in present work, since simulations were performed with RANS and with the assumption of steady flow, not allowing for the computation of vortex instabilities and fluctuations. These features can be addressed only by using more demanding unsteady LES or DES approaches which are able (e.g., [46]) to predict the vortex breakup in the propeller which was experimentally observed in [19].

## 9. Conclusions

The paper deals with the analysis of the flow features around two ducted propulsors. In particular, two different propellers were considered, with different duct configurations: one equipped with a decelerating duct, the second with an accelerating duct.

The problem was addressed with a viscous flow solver. The capabilities of the numerical approach to deal with this test case were analyzed, comparing calculations with measurements in terms of both propeller forces and predicted wake fields.

In doing this, an automatic mesh refinement technique was tuned and applied in order to track the vortical structures generated by the propeller in the slipstream without performing overly demanding simulations. A sensitivity analysis, based on Richardson extrapolation, was carried out to assess the dependence of the solution by the mesh arrangement. This analysis showed a substantial independence of the predicted forces by the mesh.

Comparing calculations with the experimental data in terms of thrust and torque, a fair agreement was observed when the non-ducted configurations were considered. Instead, in the case of ducted configurations, some non-negligible discrepancies occurred, even if results are deemed satisfactory. In addition, further aspects concerning the propeller setup were analyzed—namely, the shaft configuration (pulling or pushing) and the effect of the boss cap shape. According to numerical results, the effects of the shaft configuration were non-negligible, while the shape of the boss cap (here analyzed only in the pushing configuration) had practically no effects on the computed forces.

The viscous code showed quite a good agreement with LDV measurements in terms of predicted wake shape and vortex positions. In contrast, some discrepancies were observed regarding the radial velocity distributions, probably due to an inaccurate solution of the blade load near the tip. This issue is stressed in the ducted configurations, which were characterized by larger differences between the computed and measured velocity distributions, particularly regarding the wake of the duct. These discrepancies may be also responsible for the small differences previously mentioned in terms of delivered forces. These aspects could be related to the limits of the RANS approach, and to its relatively crude approximate modeling of the turbulence. Finally, the generated vortical structures were analyzed (with and without the duct), showing that the presence of the duct generated secondary vortexes, overall well-captured by the calculations. The evolution of these secondary vortexes and their interaction with the main tip vortex was qualitatively described, highlighting possible consequences for propeller cavitation and noise. In conclusion, although some discrepancies with experimental results were observed, this work demonstrated the capabilities of viscous-based CFD codes to reliably predict the flow dynamics in the near and far field of an unconventional propeller system, making these tools reliable for many engineering analyses. On the other hand, if a more accurate insight into complex flow phenomena is of interest, the adoption of advanced methods is needed, considering for instance fully resolved boundary layers or the solution of the turbulence dynamics.

**Author Contributions:** Conceptualization and methodology: D.V., G.T., S.G.; Data investigation and formal analysis: D.V., G.T.; Writing, review, and editing: D.V., S.G., G.T., M.V.; Supervision: M.V. All authors have read and agreed to the published version of the manuscript.

**Funding:** The tunnel test measurements presented in this paper were partly carried out in the framework of an industrial project on the design of high-performance ducted propellers founded by CETENA S.p.A. and FINCANTIERI S.p.A.

**Conflicts of Interest:** The authors declare no conflicts of interest.

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
