# Peer review of "Numerical and Experimental Comparison of Ducted and Non-Ducted Propellers"

_jmse, doi:10.3390/jmse8040257_

Round 1

Reviewer 1 Report

General Comments

The paper is of interest for providing a direct comparison of RANS
computations with LDV measurements of the flow field highlighting
where there are disicrepancies. Most of my comments pertain to the
presentation rather than to the results themselves.

In general, figures are placed too early. The reader is constantly
seeing figures for which there is no explanation until a page or two
later in the text.

Specific Comments

Table 1: Were either of the ducts one of the standard shapes? The
accelerating duct looks very similar to a 19A duct.

Section 2: What were the dimensions of the flow domain? Did they model
the dimensions of the tunnel in the experiments? How far upstream and
downstream did they extend? How far downstream did the shaft extend?
What boundary conditions were used on the outer boundaries?

Equation 1: the variables in this equation should be described.

Figures 3 and 4: Though the intent of these figures is clear, I don't
understand the labels for the curves. What are Ktref, Kqref and
etaOref?

Section 5: Lines 207 to 214: When the shaft is reversed, the boundary
layer on the hub becomes important so it is essential to know how far
the shaft extends. This is never reported.

Figure 9: It would be clearer if only four colour levels were used
corresponding to 0, 1, 2 and 3 refinements.

Figures 10 to 14: These figures should be shown in the same order that
they are discussed in the text. Moreover, Figure 12 occurs before
Figure 11.

Figures 12 to 14: The labels on the figures are too small. It would
be better to have a single legend at the right which could be made
large enough that it is readable.

Figure 14: The captions under the sub-figures are difficult to read
because they almost run into one another. It would be better to make
them a bit narrower.

Line 306: Is this meant to refer to Figure 15? Figure 15 is never
mentioned in the text. However, if this should be Figure 15, then
Figure 16 would never be mentioned in the text.

Figure 16: I find this figure unclear. Is it supposed to be showing
that the pitch of the vortices is not changing? The green lines
should be described in the caption.

Typos/Grammar

Line 4: are not able to insight -> are not able to provide insight

Line 17: principle -> principles

Line 18: contest -> context

Line 22: "accounted" is not the correct but I'm not sure what is
intended. Perhaps "encountered"?

Line 24: summed to the propeller thrust itself -> added to the
propeller thrust

Line 36: since -> for

Line 49: misplaced comma; spurious end parenthesis; semi-colon should
be a comma

Line 57: delete "an"

Line 61: propeller and consequently -> propeller, consequently

Line 64: its -> their

Line 88: computational -> computationally

Line 89: vortices -> vortex

Line 90: by -> of

Line 91: stationariety -> stationarity

Line 100: problems -> problem; conditions -> condition

Line 104: suit -> suite

Line 110: computations -> computation

Line 111: accordingly -> according

Line 127: in towing tank -> "in a towing tank" or "in the towing tank"

Line 140: had been -> was

Line 155: height -> eight

Line 159: contantly -> constant

Line 175: smaller range values -> small range of values

Line 177: simulations -> simulation

Line 180: propellers -> propeller

Line 181: propellers -> propeller; ducts -> duct

Lines 188, 189: obtained, changing -> obtained by changing

Line 197: pointing out -> to point out

Line 235: respect -> with respect

Line 237: respect -> with respect

Line 250: filed -> field

Line 252: change -> changes

Line 272: filed -> field

Figures 12 and 13 caption: no-dimensional -> non-dimensional

Figure 14 caption: non dimensional -> non-dimensional

Line 301: form -> from

Line 335: structure -> structures

Line 353: generate -> generated

Line 354: from -> on

Line 377: trustworthiness -> trustworthy

Line 473: spurious comma at the beginning of the line

Author Response

Reviewer 1

The authors are glad for the precise review made by the reviewer. The paper has been amended consequently. The major revisions have been highlighted in red in the text to simplify the revision process.

General Comments

The paper is of interest for providing a direct comparison of RANS computations with LDV measurements of the flow field highlighting where there are disicrepancies. Most of my comments pertain to the presentation rather than to the results themselves.

In general, figures are placed too early. The reader is constantly seeing figures for which there is no explanation until a page or two later in the text.

We apologize for this issue. This is mainly due to the selected layout in Latex. We have rearranged their positions for clarity, however, we are confident that when the final version (with the correct layout) is generated, everything will be finer.

Specific Comments

Table 1: Were either of the ducts one of the standard shapes? The accelerating duct looks very similar to a 19A duct. Yes, the accelerating one is the 19A duct shape. We include this sentence to avoid further doubts.

Section 2: What were the dimensions of the flow domain? Did they model the dimensions of the tunnel in the experiments? How far upstream and downstream did they extend? How far downstream did the shaft extend? What boundary conditions were used on the outer boundaries? A cylindrical domain sector has been here considered. The domain has a radius of 10 times the propeller diameter, a longitudinal extension upstream and downstream the propeller plane 5 and 10 times the diameter. The shaft is always as long as the domain; therefore, it intersects the inlet or outlet boundaries, depending on the configuration. The typical boundary conditions for this type of simulations are used: prescribing the velocity at the inlet and the pressure at the outlet; the far-field boundary adopt a symmetry plane condition.

Equation 1: the variables in this equation should be described. Some explanations have been added in the revised version.

Figures 3 and 4: Though the intent of these figures is clear, I don't understand the labels for the curves. What are Ktref, Kqref and etaOref? Due to industrial reasons, all the results are made non-dimensional with respect to reference values. The subscript “REF” are included to highlight this masking of the data.

Section 5: Lines 207 to 214: When the shaft is reversed, the boundary layer on the hub becomes important so it is essential to know how far the shaft extends. This is never reported. A sentence has been included in the revised version of the paper to highlight that the shaft line has always been considered infinite.

Figure 9: It would be clearer if only four colour levels were used corresponding to 0, 1, 2 and 3 refinements. Yes, this was the intent of the authors. A sentence has been anyway added in the revised version.

Figures 10 to 14: These figures should be shown in the same order that they are discussed in the text. Moreover, Figure 12 occurs before Figure 11. Everything has been amended in the present form.

Figures 12 to 14: The labels on the figures are too small. It would be better to have a single legend at the right which could be made large enough that it is readable. The figure has been re-generated increasing the text font size, and using a single legend out of each figure.

Figure 14: The captions under the sub-figures are difficult to read because they almost run into one another. It would be better to make them a bit narrower. The caption has been repositioned to be more clear.

Line 306: Is this meant to refer to Figure 15? Figure 15 is never mentioned in the text. However, if this should be Figure 15, then Figure 16 would never be mentioned in the text. Sorry for the misunderstanding. Everything has been corrected.

Figure 16: I find this figure unclear. Is it supposed to be showing that the pitch of the vortices is not changing? The green lines should be described in the caption. These lines in the figure have been commented in the text.

Typos/Grammar

Line 4: are not able to insight -> are not able to provide insight

Line 17: principle -> principles

Line 18: contest -> context

Line 22: "accounted" is not the correct but I'm not sure what is
intended. Perhaps "encountered"?

Line 24: summed to the propeller thrust itself -> added to the
propeller thrust

Line 36: since -> for

Line 49: misplaced comma; spurious end parenthesis; semi-colon should
be a comma

Line 57: delete "an"

Line 61: propeller and consequently -> propeller, consequently

Line 64: its -> their

Line 88: computational -> computationally

Line 89: vortices -> vortex

Line 90: by -> of

Line 91: stationariety -> stationarity

Line 100: problems -> problem; conditions -> condition

Line 104: suit -> suite

Line 110: computations -> computation

Line 111: accordingly -> according

Line 127: in towing tank -> "in a towing tank" or "in the towing tank"

Line 140: had been -> was

Line 155: height -> eight

Line 159: contantly -> constant

Line 175: smaller range values -> small range of values

Line 177: simulations -> simulation

Line 180: propellers -> propeller

Line 181: propellers -> propeller; ducts -> duct

Lines 188, 189: obtained, changing -> obtained by changing

Line 197: pointing out -> to point out

Line 235: respect -> with respect

Line 237: respect -> with respect

Line 250: filed -> field

Line 252: change -> changes

Line 272: filed -> field

Figures 12 and 13 caption: no-dimensional -> non-dimensional

Figure 14 caption: non dimensional -> non-dimensional

Line 301: form -> from

Line 335: structure -> structures

Line 353: generate -> generated

Line 354: from -> on

Line 377: trustworthiness -> trustworthy

Line 473: spurious comma at the beginning of the line

Thank you very much for the precise corrections. They have a significant impact on the overall quality of the manuscript. Your contribution has been priceless.

Reviewer 2 Report

See attached pdf file.

Author Response

Reviewer 2

Thank you for the review. All the possible source of misunderstandings, highlighted by reviewer, have been corrected, this should increases significantly the comprehension of the paper. All the revisions have been highlighted in red in the text to simplify the revision process.

In this paper, the CFD simulations of ducted and non-ducted propellers are presented and compared with EFD results. The CFD computation is based on a commercial software StarCCM+V11. The simulations are 3D, in complex geometry using AMR technique. The authors presented detailed description of the numerical and experimental set ups. The CFD results agree with the EFD results very well. The authors also discussed the discrepancies between simulation and experiments. The results presented in the paper provides a validation test for the CFD for propeller simulation. In my opinion, this paper presents useful information for researchers in various fields such as computational science, engineers and experimentalists, so I do recommend this paper for publication with some minor clarification.

I only a few minor questions and comments:

1. I suggest to write out the full names of CFD and EFD. CFD stands for Computational Fluid Dynamics and EFD stands for Experimental Fluid Dynamics? The acronyms have been replaced by their definitions in the abstract to avoid confusion. Then, the meaning of the acronym has been added in correspondence to their first occurrence.

2. AMR is the abbreviation of Adaptive Mesh Refinement, not Automatic Mesh Refinement, to my knowledge. Sorry for the misunderstanding, it has been amended in the whole paper.

3. The Figure 5 needs some clarification. On Line 183 to 185 of the manuscript, the author said:

All these simulations adopt a standard mesh set-up commonly adopted in literature which consists of about 1.5 million cells for a single blade. The denser mesh is clustered in the near field blade region as can be seen in figure 5.”

My understanding is that all three meshes in figure 5 consist of about 1.5 million cells. By zooming in, it seems that the difference between these meshes are the overall cell ratio, not near the meshes near the blade. Below I put lower-left corners of figure 5a and 5c next two each other. From the comparison of the cells in the two figures, we see that their ratio (dx in coarse mesh vs dx in finer mesh) is about 3:2. This implies that the ratio of the numbers of cells in the coarser mesh vs the finer mesh is 23: 33 =8:27. If the coarser mesh has 1.5 million cells, then the finer mesh should have 27/8*1.5 = 5 million cells, so I don’t think that the three meshes have about the same numbers of cells. Based on this, the mesh is denser in the near field of the blade because the overall resolution is higher. A better way of testing the AMR is to use the same cell away from the propeller and only refine the region near the propeller.

Sorry for the misunderstanding. The reference mesh, which as about 1.5 million cells, has been used for the propeller performance predictions. The sentence that you mention refers to this mesh (second mesh in figure 5, the reference in the text has been updated). On the contrary, figure 5 shows the three meshes adopted for the Richardson extrapolation procedure, therefore, their cells count are reported in the following paragraph. A sentence and a reference to the figure have been included in the revised paper version, to clear this aspect.

4. Most of the CFD results agree with the EFD results very well. Only in Figures 13 and 14, there are some differences on the velocity profiles, which is likely related to the boundary condition. I don’t know if there are options on different boundary conditions in the software, such as non-slip vs slip boundaries, outflow, Dirichlet or Neumann boundaries, etc. If so, the authors may want to try them and see the boundary effect.

No particular options can be selected for a wall boundary. The only possibility is to adopt a Y+ lower than 1, so avoiding the classical wall-function model. The consequence of that is decreasing the first wall cell size and consequently increasing the overall number of cells near the propeller blade. This approach unavoidably increases several times the computational costs, so it lies outside the aim of the paper.

Round 2

Reviewer 1 Report

I am happy with the paper in its revised form.

Reviewer 2 Report

The reviewer's questions have been addressed by the authors in the revised version. Recommend for publication.